# Pretraining with Masked Backstories in a Toy World

**Sultan Daniels, Dylan Davis, Gireeja Ranade, Anant Sahai**
Department of Electrical Engineering and Computer Sciences
University of California, Berkeley
Berkeley, CA 94720, USA
{sultan_daniels,dylanjd,gireeja,asahai}@berkeley.edu

## ABSTRACT

Context-enhanced learning (CEL) involves augmenting context in large language models (LLMs) with special masked context to accelerate learning. It has previously only been explored using LLMs with billions of parameters during fine-tuning because CEL requires in-context-learning (ICL) abilities to work. Here, we leverage a toy world (symbolically labeled randomly interleaved vector time-series from linear deterministic dynamical systems) that admits LLM-style "next token" pretraining and has been shown to exhibit multiple emergences of different ICL/recall abilities in tiny transformer models with mere millions of parameters.

In this toy world, we can see a late transition from ICL to in-weights learning that also corresponds to a degradation of ICL performance on time-series from systems never seen during training. We enhance pretraining with additional masked context that allows the model to make near perfect predictions on the original training examples. Masking this additional context disincentivizes the model from memorizing it, and the capability of perfect prediction on the training example disincentivizes the model from memorizing the remaining portion of the training example. Not only does this enhancement suppress in-weights learning of the specific training systems, but we also show that it improves the quality of ICL in the model, including the seemingly unrelated task of associative recall. Even more surprisingly, another experiment shows that despite such a model during training only seeing losses (and hence gradients) for tokens that are perfectly predictable, it generalizes well at test time when predicting tokens that are not perfectly predictable, nearly matching the performance of the optimal solution for those cases.

## 1 INTRODUCTION

The machine learning community is used to the general idea that models get better with scale. We carve out an exception for "overfitting/overtraining" but reality is more complicated. The phenomenon of "inverse scaling" exists (Lin et al., 2022; Michaelov & Bergen, 2023) and McKenzie et al. (2024) showed that some capabilities of language models may get worse (like subsequent fine-tunability as in Springer et al. (2025)) as model size and training compute grows[1], although many other capabilities are getting better[2]. An example of an inverse scaling task is prompting a model to always complete with the word "heavy" and then giving it "Absence makes the heart grow " — larger models are more likely to disregard the instructions and respond "fonder." This classic example illustrates a general pattern — inverse-scaling examples seem to manifest a tension between what's been learned in the weights (IWL) and what should be learned from the context (ICL).

---

[1]Wei et al. (2023); Wu & Lo (2024) add further nuance to the inverse scaling discussion, by showing that at even larger scales, there can be "double descent" (Nakkiran et al., 2021) improvement even after some worsening in the inverse-scaling capabilities.

[2]This complex dynamics of the LLM capability evolution necessitates a large repertoire of methods for evaluating, controlling, and modulating the progress of language models at a finer-grained capability by capability level. Currently, model merging, data curation and context distillation, and post-training techniques are some methods that try to mitigate this problem. See Appendix A for more.

In this paper, we study an algorithmic toy problem, where small scale models (millions of parameters) exhibit both IWL and ICL *as well as* the phenomenon of some capabilities getting better while others get worse at some point in training. We investigate a technique for modulating the development of a capability during training inspired by context-enhanced learning (Zhu et al., 2025) and gradient starvation (Tachet et al., 2020; Pezeshki et al., 2021). The approach novelty in our work is twofold: (a) Our toy problem exhibits the "some abilities get better while others get worse" during training — earlier works studying ICL/IWL transitions did not have toys that showed both directions simultaneously; and (b) we engage with an intervention of "masked context" which is qualitatively and substantively distinct from the kinds of regularization or data-mix curation interventions studied in earlier toys. We show that the intervention works in blocking the transition to IWL, ends up strengthening ICL including a seemingly unrelated aspect of associative recall, and even more surprisingly, suggests that there is something about the inductive bias of these architectures that lets them successfully extrapolate to a task that they've never seen training gradients for.

## 2  SETUP

Just as in Daniels et al. (2025), we build off of Garg et al. (2022); Sander et al. (2024) and focus on the orthogonally evolved system family, where the system is defined by $U \in \mathbb{R}^{5 \times 5}$, a random orthogonal matrix. Each $U$ is generated (Mezzadri, 2006) to ensure a uniform sampling over all $\mathbb{R}^{5 \times 5}$ orthogonal matrices. The initial state is $\mathbf{x}_0 \sim \mathcal{N}\left(0, \frac{1}{5}I\right)$, with state updates: $\mathbf{x}_{i+1} = U\mathbf{x}_i = U^{i+1}\mathbf{x}_0$. The system state is in-principle perfectly predictable, but only after six positions are observed:

$$U = \begin{bmatrix} \mathbf{x}_1 & \mathbf{x}_2 & \mathbf{x}_3 & \mathbf{x}_4 & \mathbf{x}_5 \end{bmatrix} \begin{bmatrix} \mathbf{x}_0 & \mathbf{x}_1 & \mathbf{x}_2 & \mathbf{x}_3 & \mathbf{x}_4 \end{bmatrix}^{-1}. \tag{1}$$

Following Arora et al. (2023), we add an aspect of associative recall to our toy task. To form a training trace, we interleave segments of observation sequences from a library of 40,000 orthogonal systems into a length-251 context window. The exact interleaving procedure is described in Daniels et al. (2025) and illustrated as the "original training example" in Fig. 1.

See Appendix C for details on how the model was evaluated.

## 3  TRAINING WITH MASKED BACKSTORIES

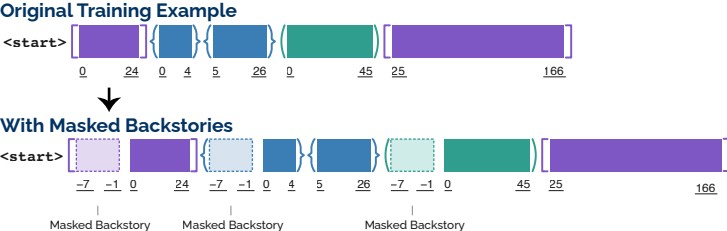

Figure 1: Training with masked backstories consists of reversing the evolution of a linear system when it appears in the training example for the first time, prepending these state observations to the corresponding segment, and masking any training loss on these backstory indices.

Whenever any particular system appears in a training example for the first time, it begins with the idiosyncratic initial state that happened to be randomly sampled for that system when building the training library as specified in Daniels et al. (2025). These early indices (from the first to the sixth) are not perfectly predictable by pure ICL alone, since 6 state observations are required to predict with an MSE of zero as shown in (1). Looking at Fig. 2a, which shows the performance on training traces, the crossed curves dip below their corresponding dotted lines towards zero squared error after more than $1 \times 10^7$ training examples have been seen by the model during training. The only way to achieve this preternatural performance is by memorizing facts rather than doing ICL.

In Fig. 2c, notice that after seeing $3 \times 10^7$ training examples, the model's error on held-out test data sharply increases above the level of the corresponding pseudoinverse predictor levels. The transition to IWL has hurt ICL performance.

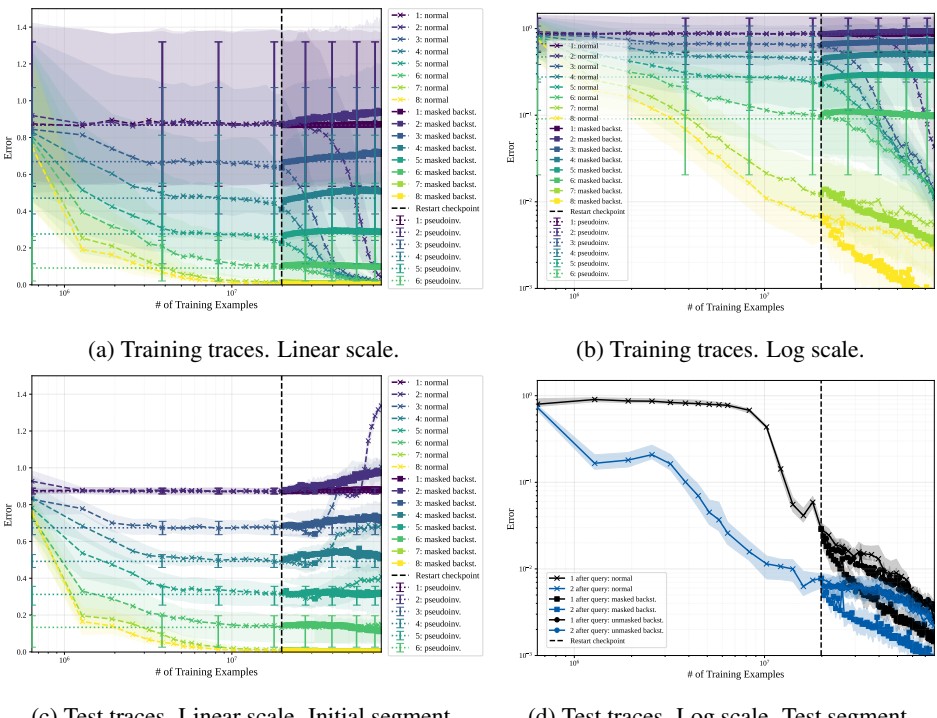

(a) Training traces. Linear scale.

(b) Training traces. Log scale.

(c) Test traces. Linear scale. Initial segment.

(d) Test traces. Log scale. Test segment.

Figure 2: The median squared error of the normally trained model (crossed curves), the masked backstory trained model (squared curves), and the pseudoinverse predictor (dotted lines). The horizontal axes measure training progress in terms of examples seen. (Multiply by 250 to get tokens)

To suppress this in-weights learning (in the style of Lampinen et al. (2025); Park et al. (2025b)), if a system $U$ appears for the first time in a training example, we evolve the system in reverse for seven time steps by successively applying $U^T$ to the initial state $\mathbf{x}_0$ to obtain $\{\mathbf{x}_{-7}, \ldots, \mathbf{x}_{-1}\}$. We call these 7 state observations a "backstory". We then prepend this backstory to the system segment as seen in Fig. 1. Finally, when the training loss is computed, we mask any losses incurred on the backstory indices. This masking is qualitatively identical to the CEL approach in Zhu et al. (2025), and so is in the spirit of leaning on the model's ICL ability.

## 3.1 RESTARTING TRAINING WITH MASKED BACKSTORIES

Our main experiment is to choose a checkpoint (at $2 \times 10^7$ training examples up to that point in training, and we call this checkpoint the "restart checkpoint") before the generalization performance has deteriorated and continue training with masked backstories from that point. We also conducted experiments where a model was always trained with masked backstories, and never received loss on any of the first 7 indices of an initial segment. This means neither IWL nor ICL were ever encouraged for these indices. Appendix F shows that these models *still* manage to make predictions that converge towards optimal on indices that never received loss.

Fig. 2a shows that on the training data, the intervention by masked backstories completely blocks and even reverses the memorization of the training data. For the most part, the curves are near the level of the optimal pseudo-inverse predictor, with some additional degradation visible for the earlier time steps. The transition to IWL appears blocked, and indeed Fig. 2c shows that on held-out test traces, the degradation in ICL performance seen in the crossed curves is largely not present in the square curves with the masked-backstory intervention. Instead, we only see the same slight degradation in the purple and blue curves corresponding to earlier time steps that we see in the training data. There's no discrepancy between the training and test.

Log-scale plots in Fig. 2b and Fig. 2d show a further benefit from the intervention. In Fig. 2b we see that the performance for later time indices (when perfect ICL reconstruction is possible) gets

even better with the masked backstory intervention (the Appendix shows this is the same with held-out data). And Fig. 2d shows that the black (and blue) curves associated with associative recall performance continue to improve, and actually improve even faster with the masked backstories.

## 4 RESULTS FROM FURTHER INTERVENTIONS

As the masked backstories intervention changes many aspects of the training algorithm, in Appendix F.2 we provide results of experiments that isolate these aspects. Fig. 3 provides a summary.

Figure 3: Results from various modifications of the masked backstories intervention. Each row corresponds to a trained model, and each column an evaluation dataset. For example, the entry in the first row and column shows that the model trained with no backstories performs preternaturally well on training data at the initial segment of the haystack.

## 5 DISCUSSION

Training models to perform next-token prediction on interleaved time-series generated from linear dynamical systems provides a controlled setting for studying training dynamics. The combination of associative recall (from interleaving) and least-squares-style ICL (from the dynamics) allowed for the toy to exhibit multiple capabilities, in which an ICL-IWL transition hurt one of them while another continued to improve during training. This simple data model also facilitated the conception of training with masked backstories, and devising experiments to validate this technique.

The observation that models trained with masked backstories throughout the entirety of training (Appendix F) are still able to make reasonable predictions for indices that were always masked opens up questions about the inductive biases of the transformer architecture. There are infinitely many possible predictors for this problem that would achieve zero MSE once six observations are seen, but would obtain worse MSE than the pseudoinverse predictor when predicting the first six indices. For example, randomly projecting each observation vector, performing the pseudoinverse prediction in the new subspace, and then projecting the predicted vector back to the original subspace. An interesting theoretical question is, why does the transformer architecture have the proclivity to find something close to the min-norm solution in original coordinates when the training objective does not require this and weight matrices are initialized randomly? Empirically, are there other tasks or scenarios where a model can incidentally learn a capability without any loss fueling its learning?

Furthermore, it is unclear exactly why the additional masked context leads to improved in-context associative recall and in-context learning on later indices in initial segments. This phenomenon closely aligns with the claims of Singh et al. (2025) where ICL and "context-constrained" IWL compete in the first attention layer of a two-layer transformer, but share circuitry in the second layer. Suppressing IWL with masked backstories could then be allowing the ICL circuit to develop stronger in the early layers of our model without the interference of IWL circuit development.

As Qi et al. (2025) showed that it is possible to automatically generate additional context for language data using another LLM (in their case GPT-4), this opens the opportunity for larger-scale experiments for language model training with masked backstories.

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

APPENDIX CONTENTS

# A    RELATED WORK

**Memorization in LLMs**   It has been observed that modern large language models sometimes memorize training examples into their weights (Xiong et al., 2025; Szep et al., 2026). This memorization is a strong instance of in-weights learning (IWL), where models make predictions based on weight updates Chan et al. (2022); Singh et al. (2023). Still, this memorization may help a language model generate factually correct text with good grammatical structure.

Table 1: Toy models of in-weights and in-context learning dynamics

| Paper | Task Dataset | Transformer Architecture | Claims |
|---|---|---|---|
| Chan et al. (2022) | Omniglot | 12 layers with MLP | Data distributions with many rare classes encourage ICL, while common repeated classes encourage IWL. ICL and IWL can coexist when the data distribution has both of these properties, like a Zipf distribution. |
| Singh et al. (2023) | Omniglot | 12 layers with MLPs | ICL can deteriorate during training. L2 regularization of MLP weights especially can mitigate ICL transience. |
| Nguyen & Reddy (2024) | Gaussian random vectors mapped to binary labels | One attention layer with a three layer MLP | IWL and ICL circuits compete, with IWL being slower to develop. ICL transience appears when attention weights are L2 regularized more than MLP weights. |
| Anand et al. (2025) | Synthetic language data for syntax probing constructed from sampled words from the Penn Treebank 3 dataset (Marcus et al., 1993) | MultiBert checkpoints (Sellam et al., 2022) | Study "structural ICL" where the model cannot use any of the semantic properties of the context tokens to make predictions. This type of ICL is also transient. A variant of active forgetting (Chen et al., 2023) which scrambles the model's token embeddings is used just at the beginning of training to mitigate the transience of ICL. |
| Park et al. (2025a) | Finite mixtures of Markov chains | Two layers with MLPs. | IWL and ICL strategies compete with each other. ICL transience is due to IWL achieving better loss on training distribution data in the long run. When the model is performing the IWL strategy, Markov chain transition matrices from the training data can be recovered from MLP weights. |
| Singh et al. (2025) | Omniglot | Two layers, attention-only | shows that "context-constrained IWL" is asymptotically preferred and in their toy setting of Omniglot classification the ICL and CIWL mechanisms shared subcircuits. They found that ICL forms faster, but is not as favorable for the model to lower training loss asymptotically while the CIWL circuit takes a while to form. Furthermore, the embryonic development of the CIWL circuit allows for ICL to emerge. They used a trick specific to the Omniglot dataset, to fix the in-context exemplar to be the same as the query exemplar and showed that this made ICL asymptotically preferred. |
| Ku et al. (2025) | Sinusoidal regression and Omniglot | Four layers with MLPs | ICL and IWL transience is task specific and depends on the relative computational costs of each strategy. For example the Omniglot classification task requires high memorization capacity but is simple for ICL, therefore IWL is slow to develop. The opposite is true for the sinusoid regression task. |
| Chan et al. (2025) | Random vector inputs mapped to random label | Two layers with MLPs | Through the perspective of expected generalization error, ICL and IWL compete for which lowers the loss more. IWL is encouraged by data that repeats frequently and ICL is encouraged by rare data that can be reasoned about through context. ICL transience is explained by the rare data showing up enough times for it finally to be memorized. |

**Methods for controlling the development of different language model capabilities**  One practical way for modulating the capabilities of deep learning models is through model merging, where the weights of multiple models are combined in certain ways to produce a single model (Yang et al., 2024). Examples include, Maiti et al. (2025) using benchmark performance of individual models to determine expert models in certain domains and computes weighted averages of their weights to achieve a high performing and robust single model, Dekoninck et al. (2024) using model merging to control the style of text generation in language models, and Ro et al. (2023) showing that training an ensemble of vision models on portions of a dataset with a variety of difficulty levels and then merging, was a more sample efficient way to obtain a single model than training on the whole dataset.

A way of suppressing a certain capability from being learned during training is suppressing tokens in the dataset that correspond to that capability (Rathi & Radford, 2026). For post-training techniques, some are using "self-distillation" to mitigate catastrophic forgetting in models by leveraging their ICL capabilities (Shenfeld et al., 2026; Hübotter et al., 2026). To enhance a certain ability when finetuning a model Snell et al. (2022) proposes to distill context. Noticing that models perform better when given instructions in context and a scratchpad for generation before predicting the final output, context distillation consists of providing a model additional context and a scratchpad along with the training examples, then recording the outputs of the model. These outputs are then used as labels for finetuning. They show that this method makes models better able to internalize the instructions than just supervised finetuning.

**Context-enhanced learning**  Context-enhanced learning corresponds to using the ICL capabilities of models to modulate their training. Similar to Snell et al. (2022) for finetuning, Qi et al. (2025) provides the training examples along with GPT-4 generated context that contains full information of the target to models, then collects their outputs to compute an "in-context editing loss" between the output conditioned on the additional context and the unconditional output. The model is then finetuned on the normal dataset, but this "in-context editing loss" is a regularization term in the training objective. They find that this process mitigates overfitting and leads to more natural outputs. Zhu et al. (2025) provides partial completions or intermediate steps towards the target of a training example to a model during finetuning and shows that models can get to better performance with exponentially fewer training examples. They also show that the additional context does not get memorized by the models. Finally, Lampinen et al. (2025) shows that ICL usually generalizes better than finetuning, so they use a model's ICL capabilities to augment finetuning datasets that lead to better generalization after a finetune.

## B  OPTIMAL PSEUDOINVERSE PREDICTOR

Following from (1), given the state observations $\{\mathbf{x}_0, \ldots, \mathbf{x}_i\}$, an optimal predictor for this problem for mean-squared error computes $\widehat{\mathbf{x}}_{i+1} = \widehat{U}\mathbf{x}_i$, where

$$\widehat{U} = \begin{bmatrix} \mathbf{x}_1 & \ldots & \mathbf{x}_i \end{bmatrix} \begin{bmatrix} \mathbf{x}_0 & \ldots & \mathbf{x}_{i-1} \end{bmatrix}^{\dagger}, \tag{2}$$

and $X^{\dagger}$ denotes the Moore-Penrose pseudoinverse of $X$. Essentially, this baseline only makes non-zero errors on the first, second, third, fourth, fifth, and sixth entry in any sequence — it gets everything else perfectly correct.

## C  NEEDLE-IN-A-HAYSTACK TESTING

To evaluate the model, we create a series of structured "needle-in-a-haystack" test traces by interleaving traces in the testing library. Each "needle-in-a-haystack" trace with $N$ systems is generated by inserting a segment of 10 state observations starting from index 0 from the testing library into the test trace (the haystack). Each of these segments is individually punctuated with a unique open and close symbol pair. We then append a query open symbol to the test trace (corresponding to the needle), followed by 10 state observations that continue the system corresponding to the query open symbol (the test segment). We call this portion of the trace the "haystack". A more detailed explanation of this procedure is in Daniels et al. (2025). See Fig. 4 for a diagram of a test trace for $N = 2$ systems in the haystack and system $U_1$ as the needle.

$$\texttt{<start>} \underbrace{\left( \mathbf{x}_0^{(1)}\ U_1\mathbf{x}_0^{(1)}\ \cdots\ U_1^9\mathbf{x}_0^{(1)} \right) \left\{ \mathbf{x}_0^{(2)}\ U_2\mathbf{x}_0^{(2)}\ \cdots\ U_2^9\mathbf{x}_0^{(2)} \right\}}\ \left( U_1^{10}\mathbf{x}_0^{(1)}\ U_1^{11}\mathbf{x}_0^{(1)}\ U_1^{12}\mathbf{x}_0^{(1)}\ \cdots\ U_1^{19}\mathbf{x}_0^{(1)} \right)$$

Query   1 After Query   2 After Query

Figure 4: Needle-in-a-haystack test example. This shows a two system haystack, where the dark gray region corresponds to the needle segment and the light gray region corresponds to the entire haystack.

## D    MODEL DETAILS

Building off of the codebase in (Du et al., 2023), which was influenced by (Garg et al., 2022), we train a 10.7M parameter GPT-2 style transformer to perform this task. Our model has hidden dimension 192, 24 layers, and 12 heads. Our model's input embedding is $192 \times 57$ dimensional. The model's output layer is $5 \times 192$ to ensure that the model makes 5-dimensional predictions. The input and output layers are untied (Du et al., 2023). Three other smaller models were also trained in Daniels et al. (2025), but we focus on the largest model in this work because it exhibits in-weights learning much earlier in training than the others.

We used a batch size of 640, a learning rate of $\approx 1.58 \times 10^{-5}$, and most experiments were trained on a single NVIDIA H200 GPU (Boerner et al., 2023). A single training run takes around 5 days. We used the AdamW optimizer (Loshchilov & Hutter, 2019) and trained using mean-squared error loss.

## E    MORE FIGURES FOR RESTARTING TRAINING WITH MASKED BACKSTORIES

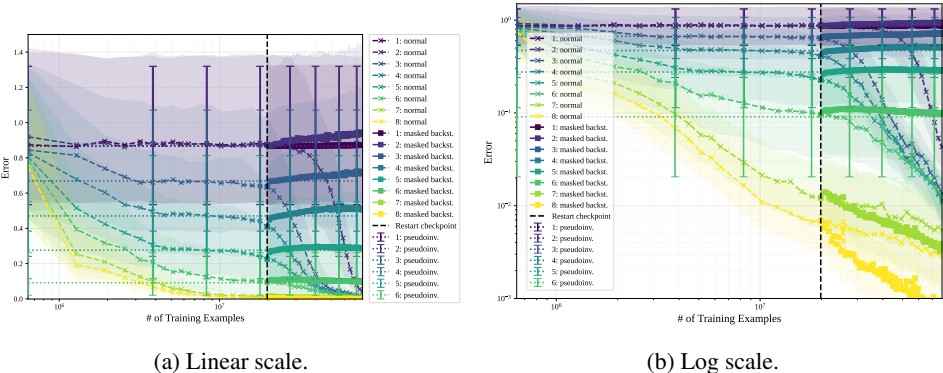

(a) Linear scale.                          (b) Log scale.

Figure 5: Median squared error of the normally trained model (crossed curves), the masked backstory trained model (squared curves), and the pseudoinverse predictor (dotted lines) on specific indices into training traces vs the number of training examples seen so far during training. The crossed curves eventually sharply decrease towards zero (signifying a transition to in-weights learning) while the squared curves do not.

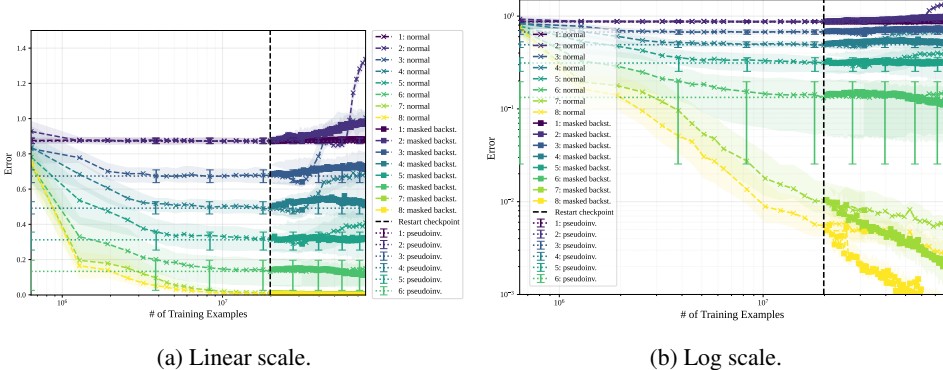

| (a) Linear scale. | (b) Log scale. |

Figure 6: Median squared error of the normally trained model (crossed curves), the masked backstory trained model (squared curves), and the pseudoinverse predictor (dotted lines) on specific indices into held-out traces vs the number of training examples seen so far during training. Notice that the crossed curves sharply increase late in training while the squared curves remain closer to their corresponding dotted lines. For a five system haystack.

## E.1 UNMASKED BACKSTORIES: TEASING APART THE EFFECT OF MASKING VS SEGMENT LENGTHS

Further analyzing this positive effect on associative recall performance, we notice that augmenting the training traces with masked backstories means that the minimum length of a segment from a system that was previously unseen in the training example is 7, while the minimum length for the original interleaving procedure is zero (see (Daniels et al., 2025)). A minimum length of 7 means that the full system is learnable by (1), and consequently, when queried for recall the model now more frequently recalls systems that it has more context about. To check if the improved performance is just due to this shift in the distribution of segment lengths in the training data, we trained a model with *unmasked* backstories. This model sees the same training data as the masked backstory trained model, the only difference is that the training loss on the backstory indices is not zeroed out. The performance of the unmasked backstory model shows us any ICL performance gains that can be made just by the shift in the segment length distribution. In Fig. 7 we see that the unmasked backstory trained model memorizes the backstories as opposed to the unbackstoried training data, and it still has a corresponding degradation in ICL on the initial haystack segment. Therefore, a shift in the length distribution alone will not remedy this effect.

In Fig. 8, the unmasked backstory model's recall performance is shown by the solid curves with triangle markers. Looking at the solid black curves for the first index into the test segment, the unmasked backstory trained model performs better than the normally trained model, but worse than the masked backstory trained model. Looking at the solid blue curves for the second index into the test segment, the unmasked backstory trained model performs at the same quality as the normally trained model, and worse than the masked backstory trained model. This shows that the improvements that we see from the masked backstory trained model cannot be solely attributed to the shift in the segment length distribution.

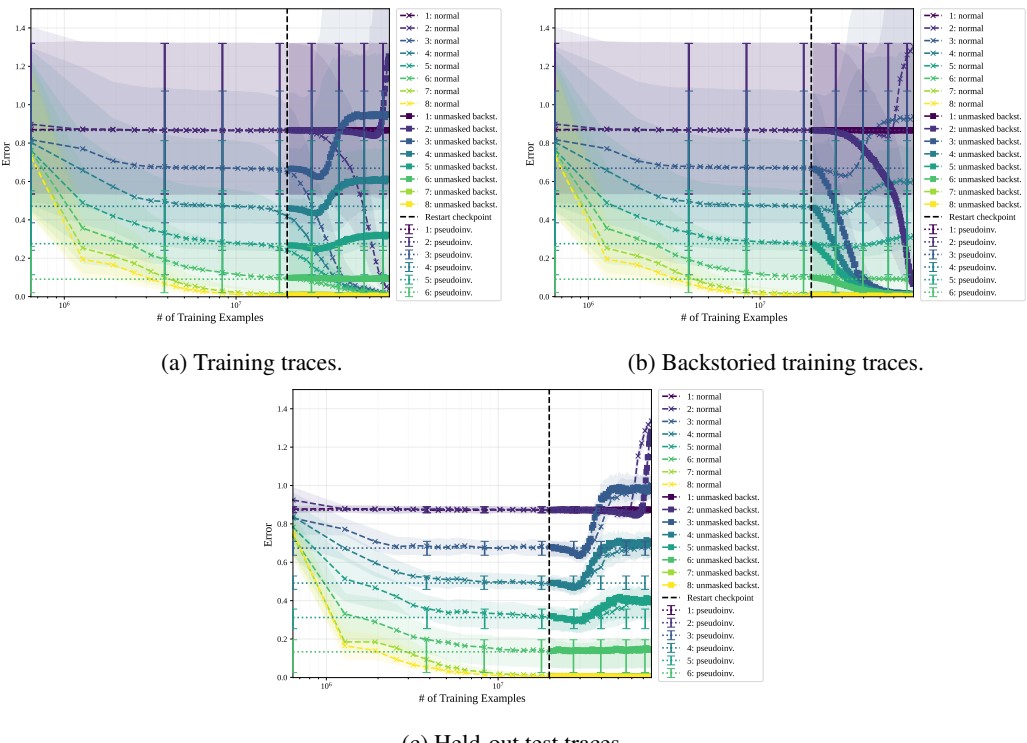

(a) Training traces.

(b) Backstoried training traces.

(c) Held-out test traces.

Figure 7: Median squared error of an unmasked-backstory trained model restarted from a mid-training checkpoint, evaluated on (a) training traces, (b) backstoried training traces, and (c) held-out test traces. Linear scale.

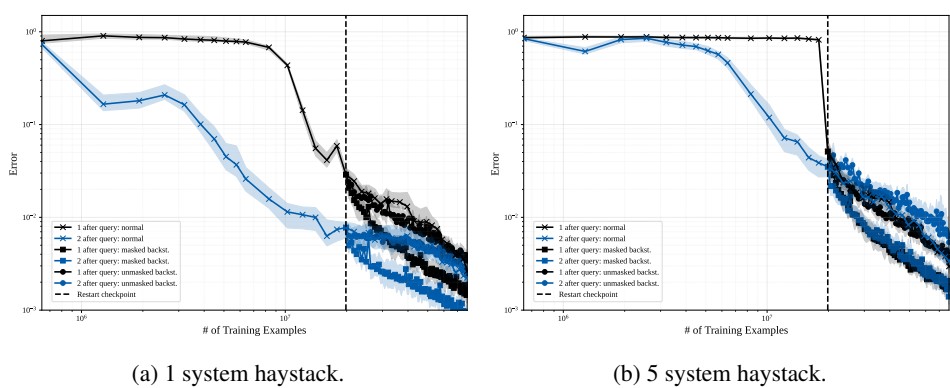

(a) 1 system haystack.

(b) 5 system haystack.

Figure 8: Median squared error of the normally trained model (dotted curves), the masked backstory trained model (diamond curves), and the unmasked backstory trained model (triangle curves) on indices one and two into the test segment of held-out "needle-in-a-haystack" traces vs the number of training examples seen so far during training. See that the masked backstory trained model's squared error (diamond curves) is always the lowest for each index.

# F  TRAINING WITH MASKED BACKSTORIES FROM THE BEGINNING

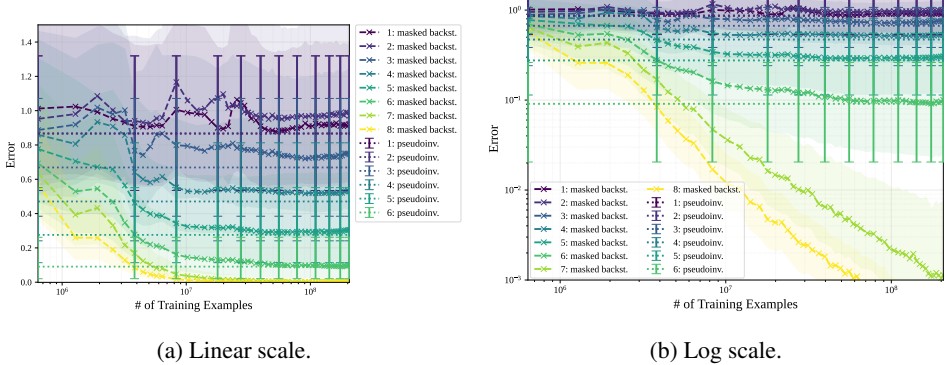

(a) Linear scale.  (b) Log scale.

Figure 9: Median squared error of the masked backstory trained model (crossed curves) and the pseudoinverse predictor (dotted lines) tested on backstoried training traces vs the number of training examples seen so far during training. The model's prediction error converges to the pseudoinverse predictor's error without any sudden shifts to in-weights learning.

Section 3.1 showed that when restarting training with masked backstories from a checkpoint that was trained using the normal interleaving procedure before the model's generalization performance deteriorates, the model's transition to in-weights learning is blocked, and the model's in-context associative recall abilities are improved. Now, we ask the question: what happens when we exclusively train a model using the masked backstories method?

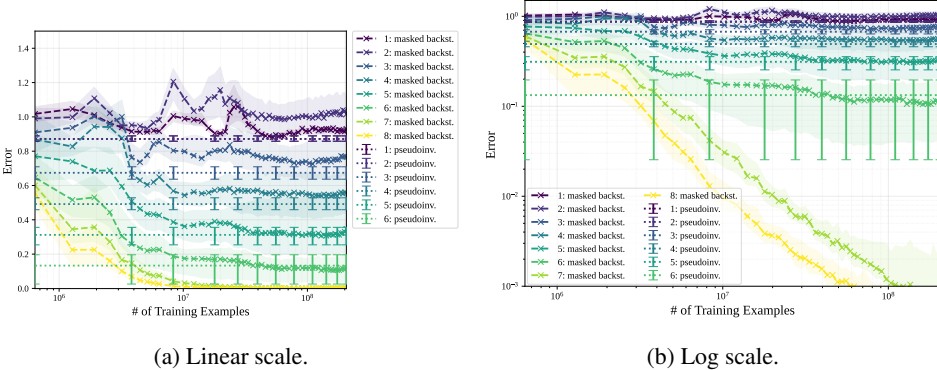

(a) Linear scale.  (b) Log scale.

Figure 10: Median squared error of the masked backstory trained model (crossed curves) and the pseudoinverse predictor (dotted lines) tested on held-out traces vs the number of training examples seen so far during training. There is no degradation in the model's generalization performance.

First, we check if the model trained using masked backstories from the very beginning exhibits a transition to in-weights learning like the normally trained model. We check this by testing this model on the training traces with prepended backstories, since it is the prepended backstories now that have irreducible loss for this model. In Fig. 9, we plot the median squared error of the pseudoinverse predictor (dotted lines) and the transformer model (dashed curves with cross markers) for the first through the eighth index into the backstoried training traces. In this figure, we see that through $2 \times 10^8$ training examples seen during training the transformer model's median squared error converges down almost to the pseudoinverse predictor's median squared error without any sudden drop down to zero median squared error as we saw with the normally trained model at around $2 \times 10^7$ training examples in Fig. 5. Additionally, we also test the model on held-out "needle-in-a-haystack" traces and Fig. 10 shows the corresponding plot for generalization performance. Notice that the crossed curves remain horizontal and do not sharply increase through $2 \times 10^8$ training examples seen unlike the crossed curves in Fig. 6 at $3 \times 10^7$ training examples seen for the normally trained model.

These two results together show us that the model trained with masked backstories model from the beginning does not transition to in-weights learning for the early indices up to this point in training.

> Since this model was trained with masked backstories from the beginning, it was never penalized during training for poor predictions on the first 7 indices of an initial segment. Nonetheless, it still developed the ability to make near optimal predictions for these indices.

The model's median prediction error approaching the pseudoinverse predictor's median prediction error in Fig. 10 provides the evidence for this. Furthermore, one could think of the task of predicting the next vector in the segment when less than 6 observations have been seen as a more difficult task than predicting the next vector when there are enough observations to know the underlying system perfectly. This is because when there are not enough observations, the model must learn how to deal with the extra degrees of freedom caused by the underdetermined system of equations. Therefore, in this toy setting, the model is exhibiting a form of "easy-to-hard" generalization (Hase et al., 2024; Sun et al., 2024; Schwarzschild et al., 2021; Song et al., 2025), where its ability to predict the next vector perfectly when there are enough observations also allows it to reasonably predict the next token when it faces uncertainty.

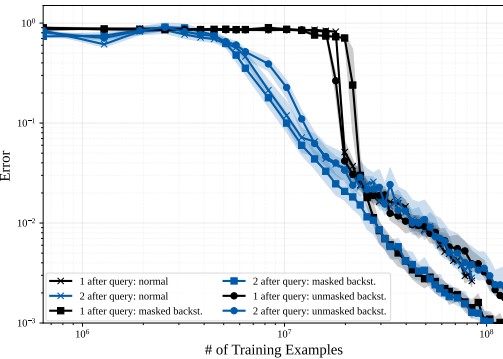

Figure 11: Median squared error of the normally trained model (dotted curves), the model trained with masked backstories from the beginning (diamond curves), and the model trained with unmasked backstories from the beginning (triangle curves) on indices one and two into the test segment of held-out "needle-in-a-haystack" traces with 5 systems in the haystack vs the number of training examples seen so far during training. Again, the masked backstory trained model's squared error (diamond curves) is always the lowest for each index.

The solid curves in Fig. 11 show the recall performance of the normally trained model (dot markers), the masked backstory trained model (diamond markers), and the unmasked backstory trained model (triangle markers). First notice that the diamond marked curves stay below the dot marked curves, which means the masked backstory trained model's associative recall ability is superior to the normally trained model's. Next looking at the triangle marked curves, in the beginning of training they follow improved performance of the masked backstory trained model's, but later in training around $3 \times 10^7$ training examples seen, the unmasked backstory trained model's improves at a slower rate than the masked backstory trained model. Training a model from scratch with masked backstories seems to lead to a decisive advantage in associative recall performance.

## F.1    PERFORMANCE AT ANOTHER HAYSTACK SEGMENT

In the main paper, we focus on first indices of a needle-in-a-haystack prompt, but in principle, memorization can occur at the beginning of any of the subsequent haystack segments as well. Here, we find that not only does the model not memorize at these later positions, its performance for predicting both training and test data degrades there as well.

In Fig. 12a, performance on the training data is shown. Memorization is clearly visible in the first haystack segment, as the dashed curves sharply drop towards zero. Contrarily, this is not true in the second haystack segment. Surprisingly, the solid curves actually rise late in training, showing

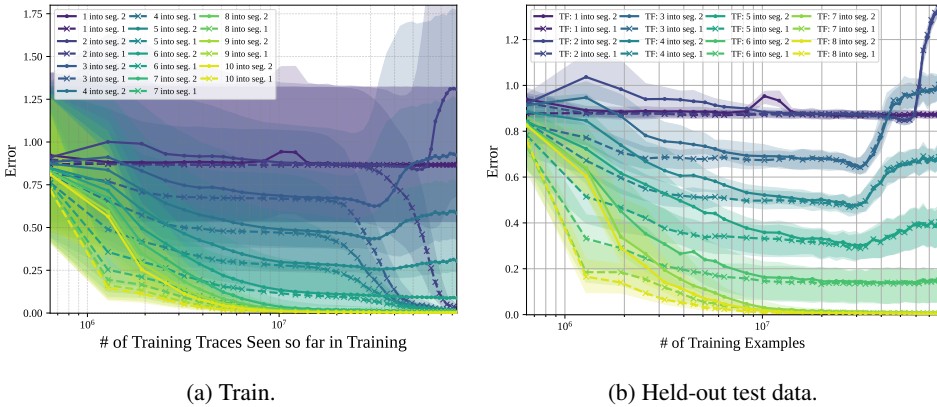

(a) Train.

(b) Held-out test data.

Figure 12: The median squared-error of the normally trained model on the first system to appear in the haystack (dashed curves with crosses), and the second system to appear in the haystack (solid curves with dots). In Fig. 12a, memorization is visible in the first haystack segment, but this is not true in the second haystack segment. Notice the degradation in the error for most indices. In Fig. 12b there is degradation for both the initial haystack segment and the subsequent one.

that the development of memorization in the initial segment leads to the model becoming worse at predicting its training data in subsequent segments. In Fig. 12b, it is shown that memorization of the initial segment leads to degraded generalization performance for both segments.

This motivates our alternate intervention of only applying masked backstories to the initial haystack segment, with results shown in Section F.2.1. Nonetheless, the results in that section confirm that backstories applied to every segment where a system appears for the first time are necessary to maintain held-out test performance on the haystack segments.

Fig. 13 provides the performance on the first and second haystack segments for the masked backstory model. Here, we see that all curves, but the unpredictable first index of the second system, do not degrade and maintain their near optimal performance.

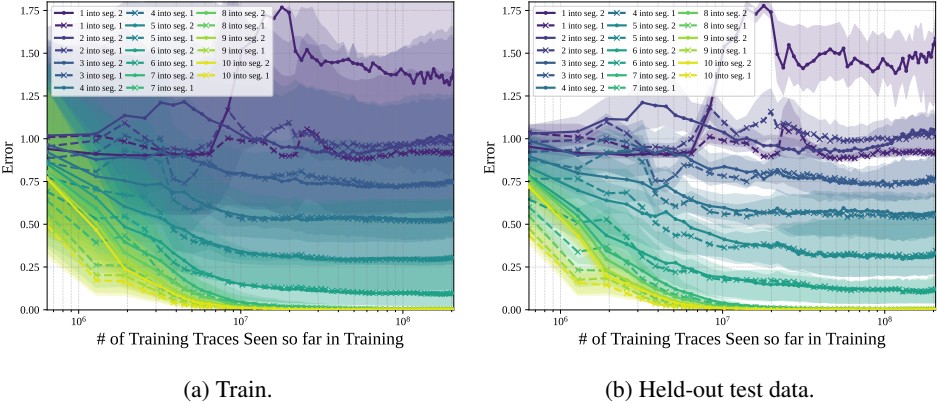

(a) Train.

(b) Held-out test data.

Figure 13: The median squared-error of the masked backstory trained model on the first system to appear in the haystack (dashed curves with crosses), and the second system to appear in the haystack (solid curves with dots). Other than the unpredictable first index in the second haystack, no other indices show degraded performance.

## F.2 FURTHER INTERVENTIONS

To better justify our design choices for pretraining with masked backstories, here we provide results for modified implementations of masked backstories. Some of these modifications are shown in

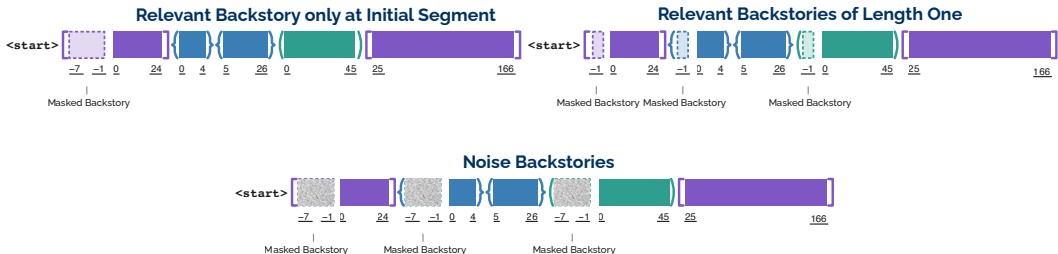

Figure 14: Diagrams for the various interventions studied in Section F.2. The backstory only at the initial segment experiment is studied in Section F.2.1, backstory of length one in Section F.2.2, and noise backstories in Section F.2.5.

Fig. 14. These results are summarized in Fig. 3 and help surface the separate effects that masked backstories pretraining has on a model's training dynamics.

### F.2.1 BACKSTORIES ONLY FOR THE INITIAL SEGMENT

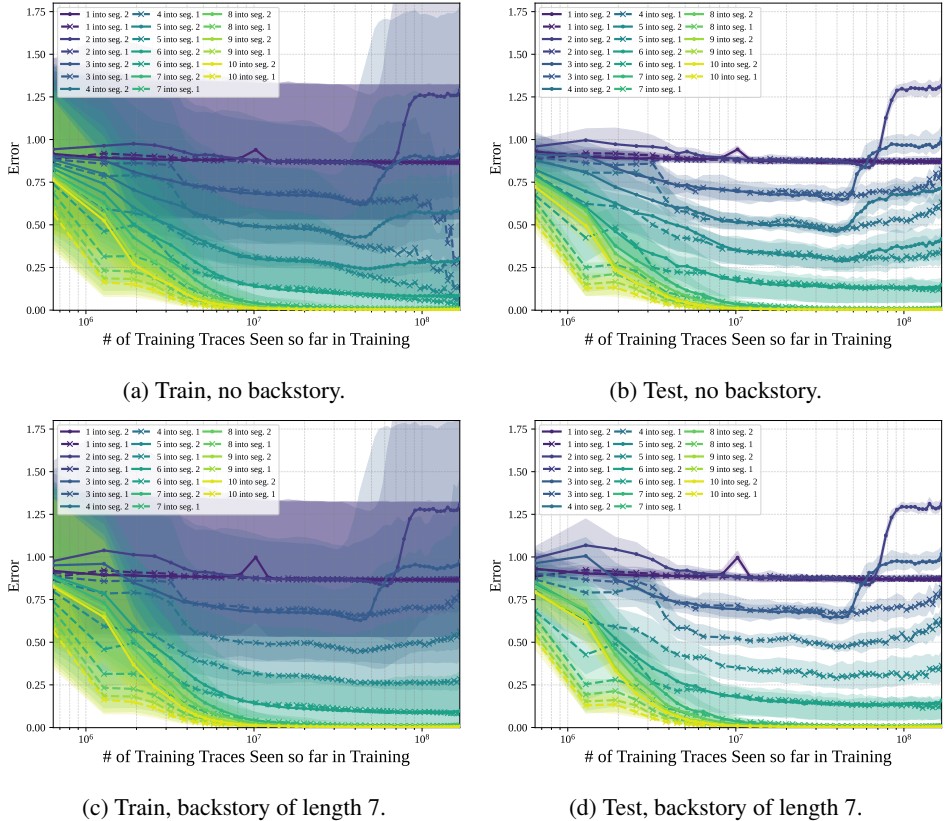

Figure 15: Training dynamics for the model trained with masked backstories only for the initial segment. Each sub-caption shows which evaluation dataset was used. Notice that Fig. 15a shows memorization of the unbackstoried training data only at the initial segment. This memorization occurs after $\approx 10^8$ training examples for this experiment while it occurs after $\approx 2 \times 10^7$ training examples for the normally trained model.

When the model is trained without any interventions, in Fig. 12 we saw that the model only memorizes the beginning indices of the initial haystack segment, and gets worse at predicting the beginning indices of subsequent haystack segments. So a natural question is, if masked backstories are only inserted before the initial segment during training, will the memorization be suppressed? Fig. 15

shows that the answer is no. Training with masked backstories only at the initial segment leads to degraded ICL abilities (Figs. 15b and 15d).

> Surprisingly, Fig. 15a shows that the model only memorizes indices in the initial segment, which the model had no gradient pressure to memorize during training, while losing performance on indices in subsequent haystack segments that did have gradient pressure during training.

This counterintuitive result shows that providing supervision at one portion of a training example, can lead to transferable abilities in other portions of the prompt and degraded abilities at the portion that received the supervision itself.

### F.2.2 BACKSTORIES OF LENGTH ONE

One of our main hypotheses for why masked backstories pretraining is an effective intervention is that the additional context allows for the model to use its in-context learning ability to predict subsequent indices that would have previously required memorization to get low pretraining loss. To test this hypothesis, we trained a model where, instead of providing a backstory of length 7, we provide a backstory of length one. One observation vector does not provide any information about the underlying system, so this backstory of length one is not additional context that can be exploited for better in-context prediction. Here, in Fig. 16a the model does not memorize the unbackstoried training data, but it does memorize the backstoried training data as shown in Fig. 16c. Again, this memorization causes degradation of performance for subsequent haystack segments and for held-out test data (Figs. 16b and 16d).

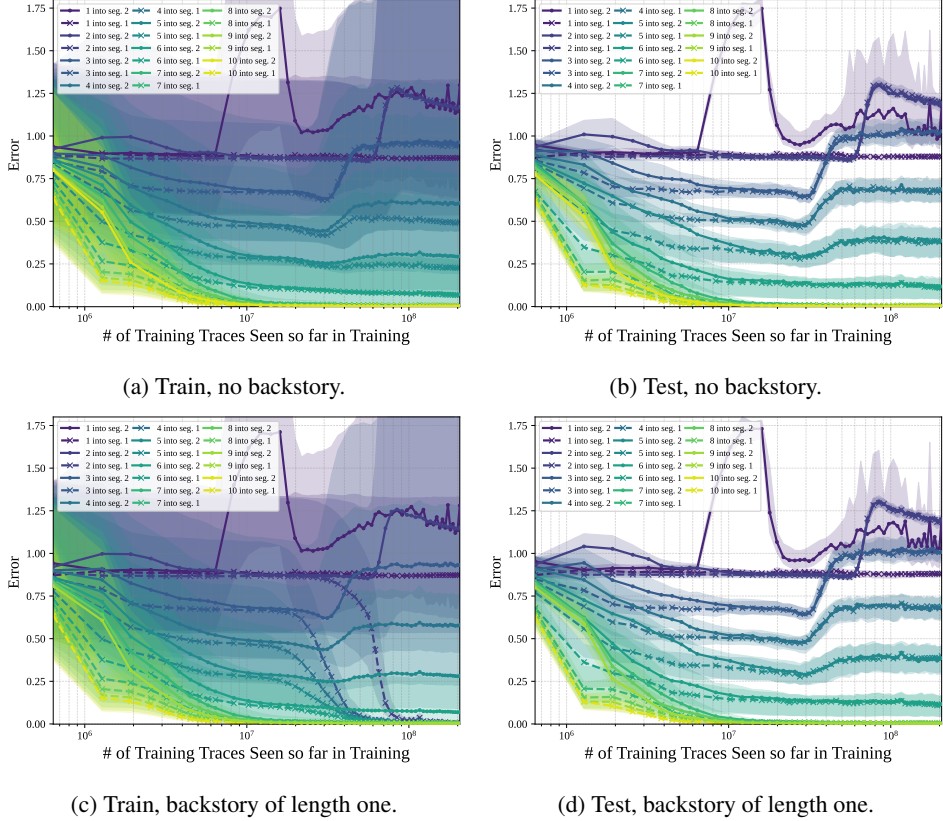

(a) Train, no backstory.

(b) Test, no backstory.

(c) Train, backstory of length one.

(d) Test, backstory of length one.

Figure 16: Training dynamics for the model trained with masked backstories of length one. Each sub-caption shows which evaluation dataset was used. Notice that Fig. 16c shows memorization of the backstoried training data, but Fig. 16a shows no memorization of the unbackstoried training data.

### F.2.3 BACKSTORIES OF LENGTH TWO

This experiment is identical to the masked backstories of length one experiment from Section F.2.2, except each masked backstory is now two indices long. With two observations, some information about the underlying system is provided. Nonetheless, the results here are the same as the one backstory case where the model does not memorize the unbackstoried training data (Fig. 17a), but it does memorize the backstoried training data (Fig. 17c).

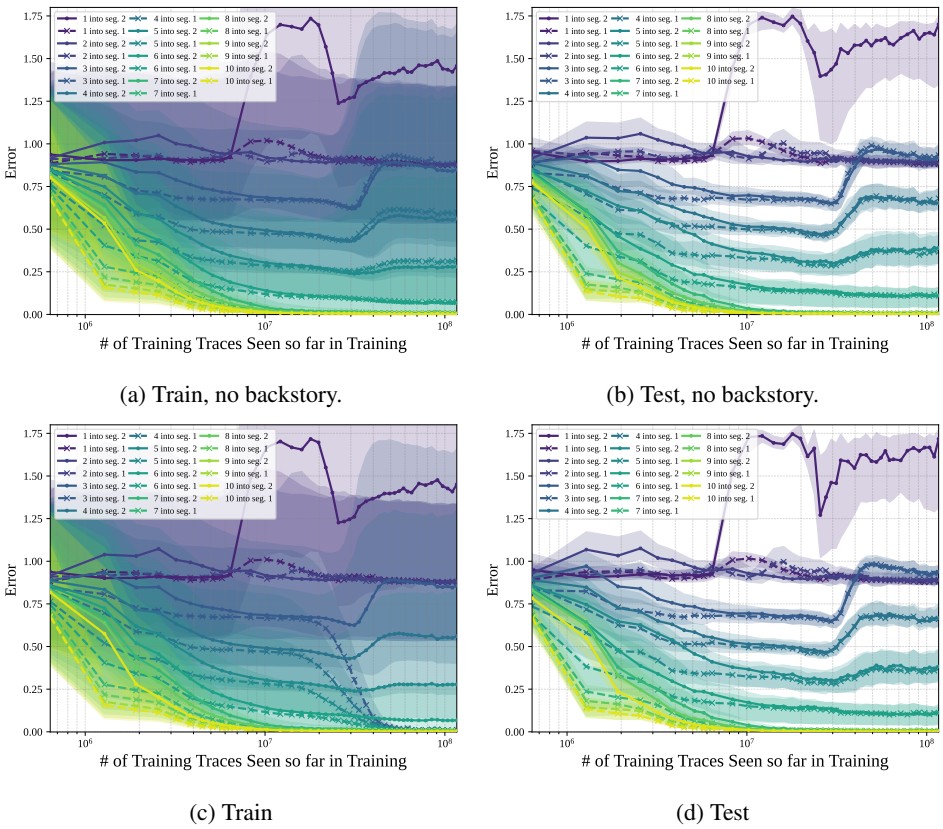

Figure 17: Training dynamics for the model trained with masked backstories of length two. Each sub-caption shows which evaluation dataset was used. Notice that Fig. 17c shows memorization of the backstoried training data, but Fig. 17a shows no memorization of the unbackstoried training data.

The results from masked backstories of length one and two experiments provide evidence that the additional context must provide significant predictive power to the model in order for masked backstories to be an effective pretraining intervention.

### F.2.4  UNMASKED BACKSTORIES

Now, we study the counterpart to Section E.1 for training with backstories from the beginning. Again, here we show that masking the backstory is an important aspect of the intervention, as the unmasked backstory trained model no longer memorizes the unbackstoried training data (Fig. 18a), but it memorizes the backstoried training data (Fig. 18c). This memorization again leads to degraded generalization performance (Figs. 18b and 18d).

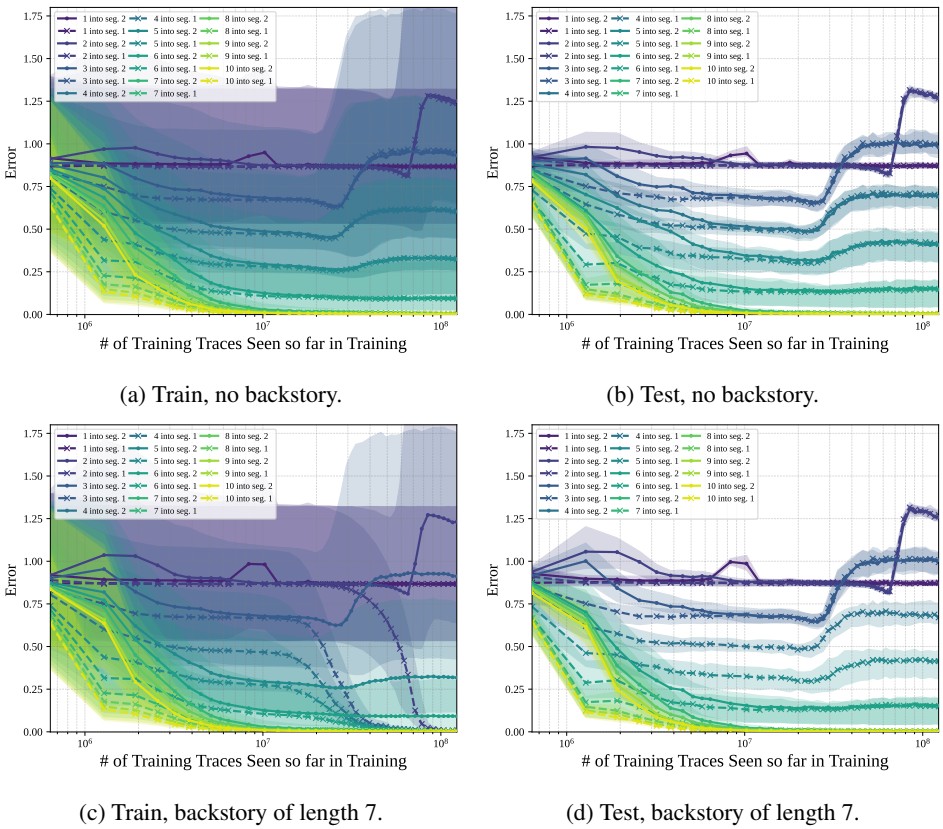

(a) Train, no backstory.

(b) Test, no backstory.

(c) Train, backstory of length 7.

(d) Test, backstory of length 7.

Figure 18: Training dynamics for the model trained with unmasked backstories of length 7. Each sub-caption shows which evaluation dataset was used. Fig. 18c shows memorization of the backstoried training data.

### F.2.5 NOISE BACKSTORIES

To further intervene on the claim that the additional context must provide significant predictive power for masked backstories to be an effective technique, we can provide 7 indices of IID Gaussian noise of the same expected squared norm as the linear dynamical system observations at the first occurrence of each system in a training example. Masking the loss of this noise then results in noise backstories. As this IID noise is unpredictable, the model must learn to ignore the noise, then make optimal prediction on the linear dynamical system observations afterward. Here, we see in Figs. 19a and 19c that the model again memorizes the first indices that are memorizable. In this case they are indices 9 and 10 in the initial haystack segment. This experiment again underscores the importance of additional context that provides information that can be used for prediction.

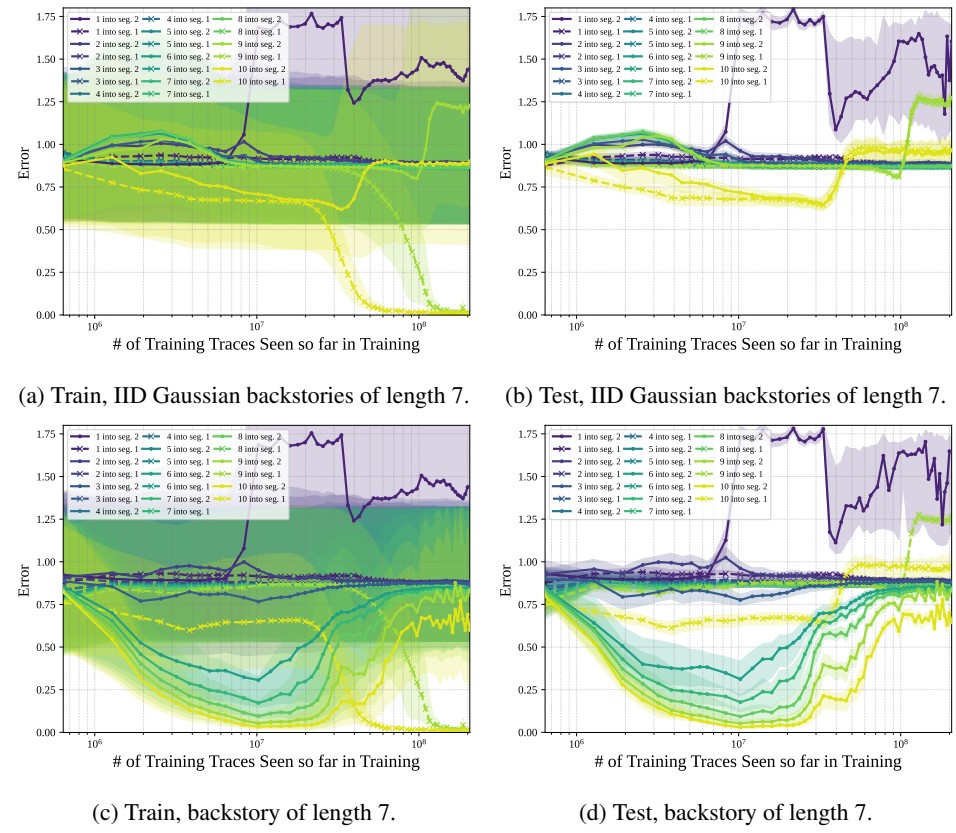

(a) Train, IID Gaussian backstories of length 7.    (b) Test, IID Gaussian backstories of length 7.

(c) Train, backstory of length 7.    (d) Test, backstory of length 7.

Figure 19: Training dynamics for the model trained with IID Gaussian backstories of length 7. Each sub-caption shows which evaluation dataset was used. Figs. 19a and 19c show memorization of indices 9 and 10 in the initial haystack segment.

Interestingly, in Figs. 19c and 19d, which show the model's performance on backstoried data from the actual linear dynamical systems, we can see that early in training the model is making accurate predictions for early indices in the second haystack segment before learning to predict near zero. This shows that the model uses in-context learning early in training before it learns that these indices in its training data consist of IID Gaussian noise.

## G    MASKING ONLY A FRACTION OF THE SYSTEMS

For all of the previous experiments, when applying masked backstories, they were applied to every training system. Now we can ask the question, if the masked backstories are only applied to a fraction of the training systems, can the ICL ability of the model be maintained while allowing for the memorization of a select group of systems?

### G.1 MASKING THREE-FOURTHS OF THE TRAINING SYSTEMS

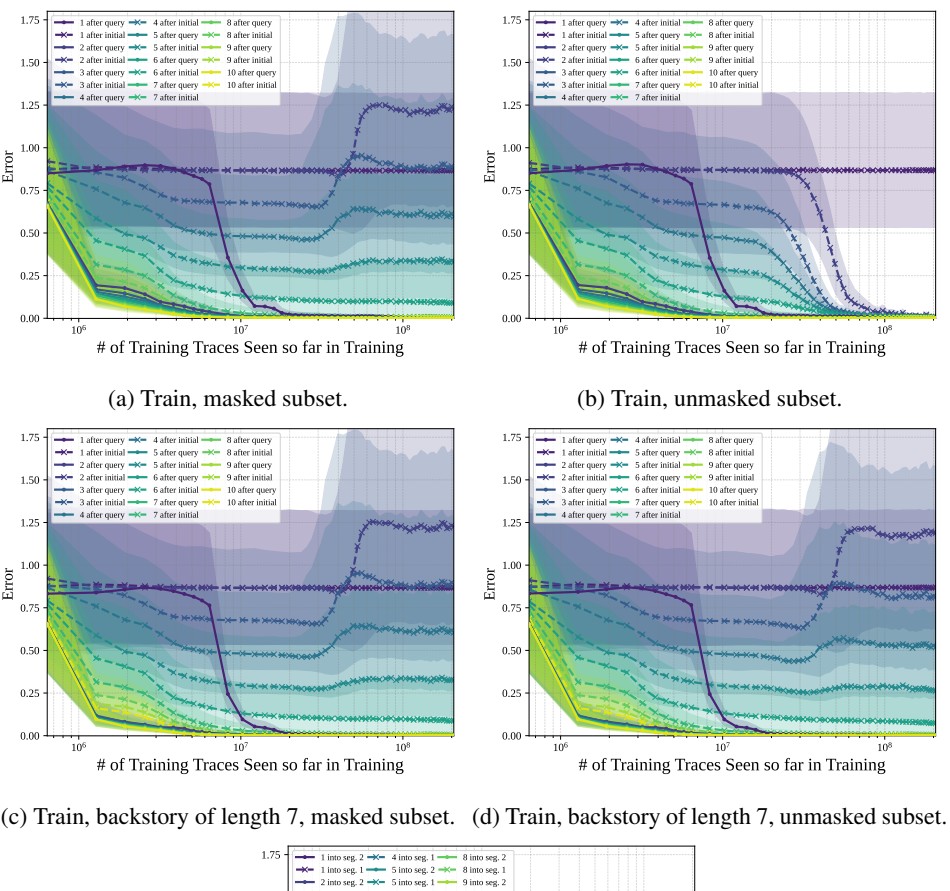

(a) Train, masked subset.

(b) Train, unmasked subset.

(c) Train, backstory of length 7, masked subset.   (d) Train, backstory of length 7, unmasked subset.

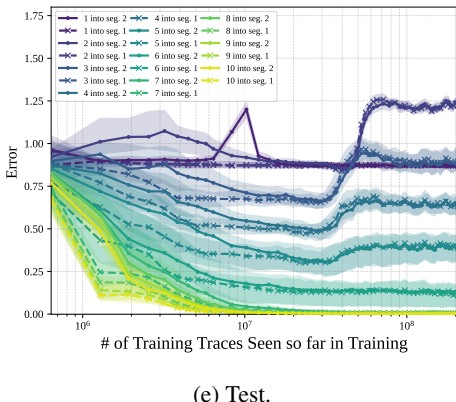

(e) Test.

Figure 20: The training dynamics of the model with 75% of the training systems masked when predicting needle-in-a-haystack style data. Each sub-caption shows which evaluation dataset was used. Fig. 20c shows that the masked systems are not memorized and Fig. 20d shows that the unmasked systems are memorized. Fig. 20e shows that the model's ICL ability is adversely affected by this memorization.

Here, we apply masked backstories to 30,000 of the training systems while allowing 10,000 of the training systems to appear in the training data without masked backstories. We provide evaluations on held-out systems, the masked training systems, and the unmasked training systems. We find that the unmasked systems get memorized after $\approx 2 \times 10^7$ training examples (Fig. 20b). The same point in training where memorization occurs in the normally trained model. For the masked systems, not only do they not get memorized, but the model's performance on this training data degrades (Fig. 20a). Overall, on held-out test data performance degrades once the 10,000 unmasked systems

get memorized as well (Fig. 20e). This shows that the model's ICL ability is very fragile to the development of in-weights memorization.

## G.2 MASKING ALL BUT TEN TRAINING SYSTEMS

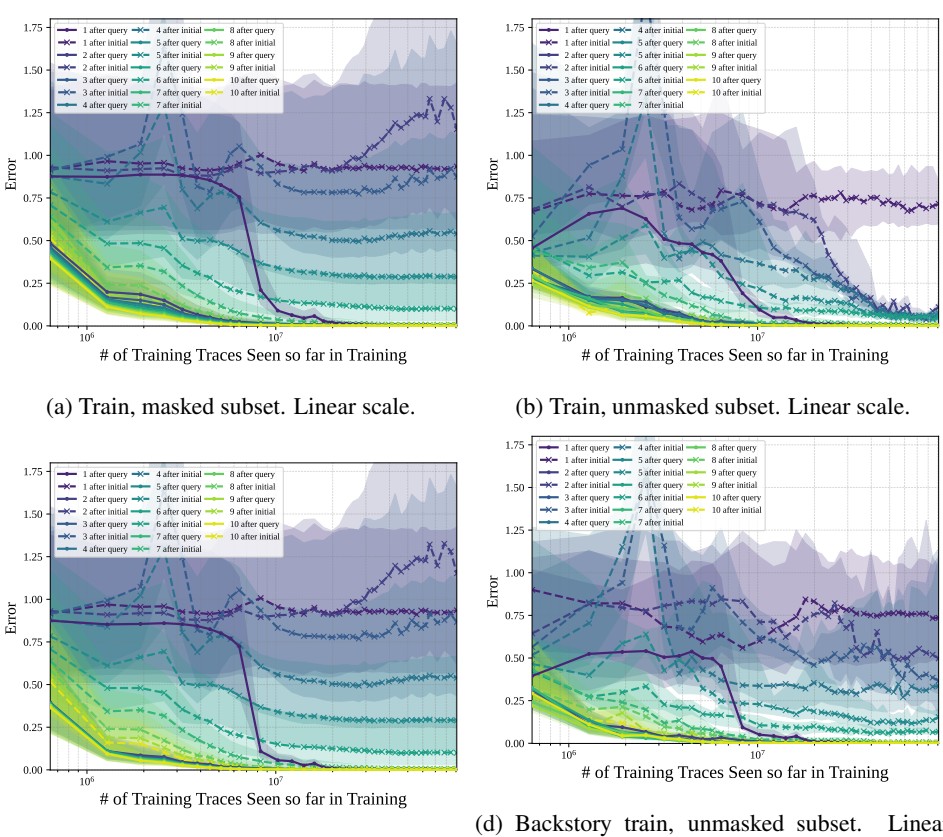

(a) Train, masked subset. Linear scale.

(b) Train, unmasked subset. Linear scale.

(c) Backstory train, masked subset. Linear scale.

(d) Backstory train, unmasked subset. Linear scale.

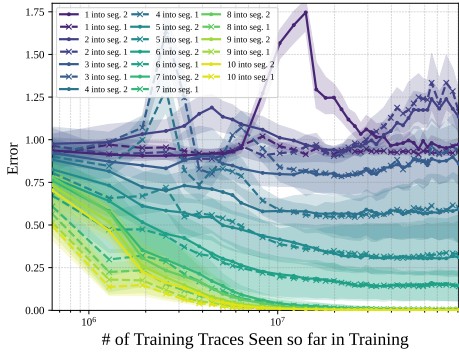

(e) Held-out test data. Linear scale.

Figure 21: The training dynamics of the model with $99.975\%$ of the training systems masked when predicting needle-in-a-haystack style data. Each sub-caption shows which evaluation dataset was used. Fig. 21c shows that the masked systems are not memorized and Fig. 21d shows that the unmasked systems are memorized. Fig. 21e shows that the model's ICL ability is still moderately affected by this memorization.

We can go to an extreme and see if the same effect from Section G.1 manifests when only 10 training systems are unmasked and 39,990 are masked. Again, in Fig. 21b we see that the unmasked systems are memorized. In Fig. 21a the masked systems are not memorized, but in Fig. 21e, the median

squared-error of the predictions in the initial haystack system for held-out test data still appears to be slowly rising. This means that even when the unmasked subset takes up $0.025\%$ of training systems, the memorization of these systems adversely affects the model's ability to perform ICL.

# H    MULTIPLE INITIAL STATE VECTORS CORRESPONDING TO A SINGLE TRAINING SYSTEM

For the previous experiments, each training system had a corresponding initial state vector. This is what allowed for the memorization of systems. A relevant question is whether increasing the number of corresponding initial state vectors to a single system will make the system harder to memorize. Here we present results for an experiment where the training library of sequences consisted of 20,000 training systems and each had two corresponding initial state vectors. This means that the training library still had 40,000 total sequences as was the case for the earlier experiments as well. Here we see that this two initial state vector to one system mapping had qualitatively no effect on the training dynamics that we see in the one initial state vector to one system setting. Memorization still occurs after $\approx 2 \times 10^7$ training examples (Fig. 22a) and ICL ability still degrades (Fig. 22b).

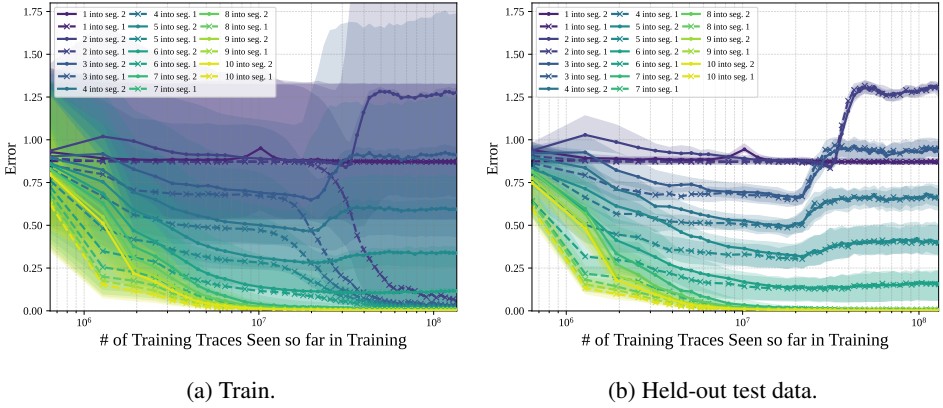

(a) Train.    (b) Held-out test data.

Figure 22: Two initial state vectors per training system. Memorization still occurs after $\approx 2 \times 10^7$ training examples (Fig. 22a) and ICL ability still degrades (Fig. 22b).

