# OpenReview forum: "Pretraining with Masked Backstories in a Toy World"
_ICLR.cc/2026/Workshop/Sci4DL — Sci4DL 2026_

### Official Review · Reviewer_DbfV · 2026-02-18

**Fit:** 3
**Significance:** 2
**Confidence:** 2

**Summary:**

The paper proposes a toy task from linear dynamical systems to study the emergence of in-context learning (ICL) and in-weight learning (IWL).
The authors show that transitioning to IWL during training leads to the degradation of ICL during testing.
The authors then mitigate this by providing additional masked context during training, which demonstrate effectiveness
in both ICL and another associative recall task during testing.

**Strengths:**

1. Investigating the emergence of ICL and IWL is of significant interest to the DL community. The authors leverage a simple task from linear dynamical system to study ICL versus IWL and their proposed intervention strategy. The methodology is sound and the design of experiments are easy to follow.
2. The paper is well written and provides lots of nice figures to explain the experimental set-up and results.

**Suggestions:**

1. The authors show that their "masked backstory" strategy for improving ICL also strengthens the "seemingly unrelated" associative recall task. However, I am not able to find the precise definition on associative recall task. Judging by Figure 8, it seems that the associative recall focuses on the token-position level loss (e.g., after the first query token, after the second query token), whereas the ICL test loss computes the average loss across all tokens in the test sequence? If so, is the associative recall truly "unrelated" to ICL? More explanations and clarifications are needed.
2. The authors mention the connection to context-enhanced learning in Zhu et al (2025). It will be nice to include more discussions, such as the possibility of extending the mathematical techniques from Zhu et al. to explain why such "masked backstory" strategy succeeds (e.g., improved gradient signals).

---

### Official Review · Reviewer_A5TF · 2026-02-25

**Fit:** 3
**Significance:** 3
**Confidence:** 2

**Summary:**

This paper introduces a novel pretraining intervention called "Masked Backstories" to address the conflict between in-context learning (ICL) and in-weights learning (IWL) within a controlled toy world with interleaved linear dynamical systems. By prepending system history and masking the training loss on those indices, the authors demonstrate that suppressing memorization (IWL) effectively blocks the degradation of generalization and even enhances associative recall performance.

**Strengths:**

1. The "Masked Backstories" approach is pretty novel and interesting, and it is effective to modulate model capabilities
2. The setup is interestingly designed such that both the improvement and worsening of different abilities simultaneously, providing a more nuanced view of ICL/IWL transitions than previous toy models
3. Extensive experiments are conducted to get robust results and a generalized conclusion

**Suggestions:**

1. The current study focuses exclusively on linear systems. Would these masked backstories facilitate ICL in non-linear or chaotic systems?
2. How would the conclusion be applied to LLM on a linguistic task rather than purely a synthetic time-series?
3. Figure 3 and other figures have very small text that is hard to read; please consider revising it for better visualization.

---

### Official Review · Reviewer_h6gZ · 2026-02-28

**Fit:** 2
**Significance:** 2
**Confidence:** 2

**Summary:**

They study a toy setting where a transformer is trained to predict trajectories from a linear dynamical system (LDS), and ask when training leads to behavior best described as in-context inference (generalizing to unseen systems from the prompt) versus in-weights learning (memorizing or encoding training-system-specific information in parameters). The central empirical claim is that standard training exhibits a late-stage transition: training error on the systems/segments seen during optimization continues to improve, while generalization to held-out systems (their proxy for ICL performance) degrades.

To mitigate this, the authors introduce masked backstories: they prepend additional reverse-time LDS state tokens, but mask the loss on those prepended tokens. The intended effect is to supply extra contextual information that can support prediction while starving gradients that would directly incentivize memorization of those backstory tokens. Empirically, they report that masked backstories can improve held-out performance and can yield accurate prediction of tokens that were never directly trained-on (loss-masked), and they interpret these results as evidence that selective loss masking can bias optimization toward an ICL-like regime rather than an IWL/memorization regime.

**Strengths:**

The main technical contribution---the masked-backstory construction---is a clean, concrete training intervention in a setting where more context versus more memorization can be probed with reasonably controlled levers (segment index, system held-out split, train vs.\ test traces). As a workshop contribution, the idea is simple enough to replicate and potentially generalize: loss-masking can be viewed as a crude mechanism for routing credit assignment away from tokens that most strongly encourage rote copying, while still letting those tokens condition the model's forward computation.

The experiments surface a real optimization pathology that is worth studying: later training can simultaneously reduce training error yet harm the particular form of generalization they care about (held-out-system prediction), consistent with a transition in the effective strategy the model uses. Even if the paper does not fully pin down mechanism, the observed tradeoff and the fact that a small change in training objective (masking) can shift the outcome is at least a plausible starting point for more careful causal and mechanistic work.

**Suggestions:**

This paper needs significant work.

The framing via inverse scaling and instruction-following failures (e.g. proverb completion) is not well matched to what is actually studied. Those examples are closer to strong prior continuation versus instruction than to the canonical ICL setup of implicit learning-from-examples, whereas the experiments are entirely in a synthetic LDS next-token prediction regime with punctuation-like delimiters. I would recommend either narrowing the paper's motivation to optimization dynamics and generalization in synthetic algorithmic/toy sequence modeling, or adding substantially stronger evidence connecting the toy phenomenon to the LLM inverse-scaling narrative.

Methodologically, the paper does not cleanly isolate why masked backstories help. The intervention changes multiple things at once: it increases context length and adds structured tokens (reverse-time states)and it changes the gradient signal via loss masking. Without ablations such as prepend backstory but do not mask it, mask an equal number of random tokens, prepend unrelated tokens with/without masking, it is hard to attribute effects specifically to masking prevents IWL rather than to distributional/architectural side-effects of extra tokens. More generally, the operationalization of ICL versus IWL is mostly via curve shapes (training error on early indices versus held-out error), which is suggestive but not diagnostic; if the goal is to argue about mechanisms/strategies, the paper needs more direct probes (e.g. representational similarity across systems, sensitivity tests to prompt perturbations, or other indicators that separate memorization from genuine in-context inference). Finally, presentation should be improved: key figures are difficult to parse, and duplicating linear/log versions in the main text adds clutter without clear benefit; one scale should be chosen for the main narrative and the other relegated to an appendix.

---

### Meta-Review · Area_Chair_sa1F · 2026-03-02

**Recommendation:** Accept

**Metareview:**

The paper studies a toy task from linear dynamical systems to investigate the emergence of in-context learning (ICL) and in-weight learning (IWL), and proposes a pre-training intervention that disincentivizes memorization and improves held-out performance. The methodology and the results are a good fit for the workshop.

---

### Decision · Program_Chairs · 2026-03-02

Accept